# Time series of Inland Surface Water Dataset in China (ISWDC) for 2000-2016 derived from MODIS archives

Shanlong Lu[1], Jin Ma[1,2], Xiaoqi Ma[1,3], Hailong Tang[1,4], Hongli Zhao[5], Muhammad Hasan Ali Baig[6]

[1]Key Laboratory of Digital Earth Science, State Key Laboratory of Remote Sensing Science, Institute of Remote Sensing and Digital Earth, Chinese Academy of Sciences, Beijing 100094, China;

[2]College of Information Science and Engineering, Shandong Agricultural University, Tai'an 271018, China;

[3]School of Earth Sciences and Resources, China University of Geosciences, Beijing 100083, China;

[4]College of Earth Science, Chengdu University of Technology, Chengdu 610059, China;

[5]State key Laboratory of Simulation and Regulation of Water Cycle in River Basin, China Institute of Water Resources and Hydropower Research, Beijing 100038, China;

[6]Institute of Geo-Information & Earth-Observation (IGEO), PMAS Arid Agriculture University Rawalpindi, Rawalpindi 46300, Pakistan.

*Correspondence to*: Shanlong Lu (lusl@radi.ac.cn)

**Abstract.** The moderate spatial resolution and high temporal resolution of the MODIS imagery make it an ideal resource for time series surface water monitoring and mapping. We used MODIS MOD09Q1 surface reflectance archive images to create an Inland Surface Water Dataset in China (ISWDC), which maps water bodies larger than 0.0625 km$^2$ within the land mass of China for the period 2000–2016, with 8-day temporal and 250 m spatial resolution. We assessed the accuracy of the ISWDC by comparing with the national land cover derived surface water data and Global Surface Water (GSW) data. The results show that the ISWDC is closely correlated with the national reference data with coefficient of determination (R$^2$) greater than 0.99 in 2000, 2005, and 2010, while the ISWDC possess very good consistency, very similar change dynamics, and similar spatial patterns in different regions with the GSW dataset. The ISWDC dataset can be used for studies on the inter-annual and seasonal variation of the surface water systems. It can also be used as reference data for verification of the other surface water dataset and as an input parameter for regional and global hydro-climatic models. The ISWDC data are

1 available at http://doi.org/10.5281/zenodo.2616035.

**1 Introduction**

Surface water is the most important source of water from planetary water resources available for the survival of both human

and ecological systems (Lu and He, 2006). It is a key component of the hydrological cycle and the key factor affecting the

sustainable development of human society and ecosystem. Both climate change and human activities have a role in affecting

the surface water availability at a given area and time. In order to locate the position and examine the change in dynamics of

the inland surface water, regional and global datasets have already been produced through remotely sensed data by various

researchers (Carroll et al., 2009; Verpoorter et al., 2014; Feng et al., 2015; Klein et al., 2014; Tulbure et al., 2016), but these

contemporary researches were limited to measuring long-term changes at high spatial and temporal resolution. Pekel et al.

(2016) quantified the changes in global surface water (GSW) over the past 32 years (1984-2015) at 30-meters resolution by

using the Landsat imagery. Klein et al. (2017) generated a 250 m daily global dataset of inland water bodies based on a

combination of MODIS Terra and Aqua daily classifications. However, the temporal resolution of the former research is near

monthly, and the latter research only produced datasets from 2013-2015 until now, while the entire MODIS archive back to

July 2002 is still ongoing (Klein et al., 2017).

In China numerous regional case studies have been done and produced some surface water datasets but only in bits and

pieces (Du et al., 2012; Lai et al., 2013; Luo et al., 2017). Their research mainly focused on lakes in the Qinghai-Tibetan

Plateau (Lu et al., 2017). Several research groups are focusing on lake water changes of this region and have produced

decadal lake surface water datasets since the 1960s (Song et al., 2014; Zhang et al., 2014, 2017; Wan et al., 2014, 2016). At

the national scale, the national wetland remote sensing datasets in 1978, 1990, 2000 and 2008 (Niu et al., 2012), the national

land cover datasets in 1990, 2000, 2010, and 2015 (Wu et al., 2017), and the national land use datasets in 1990, 1995, 2000,

2005, 2010, 2015 (Liu et al., 2018) contain the inter-decadal or 5-year time scale water surface dataset (Table 1). However,

these datasets are available with limited temporal resolution and not freely and fully shared.

<Table 1>

The most commonly used method of water extraction is based on water indices, such as the Normalized Difference Water

Index (NDWI) (Gao, 1996; McFeeters, 1996; Rogers and Kearney, 2004), the Modified Normalized Difference Water Index (MNDWI) (Xu, 2006), the Automated Water Extraction Index (AWEI) (Feyisa et al., 2014), and the Enhanced Water Index (EWI) (Wang et al., 2015). Furthermore, the single band threshold segmentation method (Li et al., 2012, Lu et al., 2017) and the multiband transformation method (Pekel et al., 2014) are also in practice. The key step for using these methods in extracting the water boundary is to determine the threshold value for segmentation. The existing threshold determination methods include human visual judgment (Huang et al., 2008; Li et al., 2012) and sample statistical analysis (Feyisa et al, 2014; Pekel etc., 2014; Pekel et al., 2016). The former relies on subjective experience, which causes the extraction results to be unstable, and thus difficult to apply on larger scales and to large volumes of data. Although the latter can get more accurate results through extensive sampling statistics, the use of a unified threshold for the whole image or whole region may produce large errors in the local area. To overcome these problems, various comprehensive classification methods are widely used. Verpoorter et al. (2014) combined the Principal Component Analysis (PCA) and the Modified Brightness Index (MBI) to generate supervised classes, and to divide these into water and non-water regions by using the decision tree method. Pekel et al. (2016) proposed an expert system by synthetic use of a visual analytical spectral library, the NDVI index, HSV transformation results, and decision tree method. Khandelwal et al. (2017) introduced a global supervised classification based approach by defining initial spatial extents of each water body, using the global sample datasets, and incorporating all the spectral reflectance bands of the MODIS imagery. Use of supervised classification or decision tree method may improve the accuracy of the water surface boundary extraction, however it increases the difficulty and efficiency of the method at the same time. Zhang et al. (2017) proposed an automatic threshold determination method based on the LBV (L, the general radiance level; B, the visible–infrared radiation balance; V, the radiance variation vector between bands) transformation of Landsat 8 OLI surface reflectance images. It was verified as an accurate, simple, and robust method for surface water extraction. However, cloud pixels and atmospheric correction influences were not considered.

China has one of the highest densities of rivers and lakes in the world. There are more than 1500 rivers with an area exceeding 1000 $km^2$ and 2928 lakes with an area larger than 1 $km^2$ which form a total surface water area of 91,020 $km^2$ (Ma et al. 2011). However, owing to the influence of climate, geography and landscape of the country, these surface water resources are unevenly distributed. They are found more in the South than in the North, and more in the East than in the

West. With the development of the economy, the increase in the demand for industrial, agricultural and domestic water has placed great pressure to these surface water systems, especially during the irrigation and drought season (Gong et al., 2011; Barnett et al., 2015). Therefore, there is an urgent need for spatio-temporal continuous surface water datasets to support the efficient and robust management of water resources, and to investigate the relationship between the national surface water and the global climate and human activities. However, until now, full public sharing data products with moderate spatial resolution and near-daily temporal resolution are still lacking in China.

In order to address these limitations and to fulfill the need to develop a comprehensive spatio-temporal dataset, this paper presents the Inland Surface Water Dataset in China (ISWDC) during the period of 2000-2016 (and will be updated continuously for the subsequent years on zenodo platform), which is derived from the 8-day and 250 m spatial resolution MODIS MOD09Q1 product. After recalling the methodology used in surface water mapping from the MODIS MOD09Q1 as described by Lu et al. (2017), the precision and accuracy of the dataset are reported, including the cross comparison with the existing national and global datasets.

**2 Study area and data**

The inland water of this dataset refers to a water body larger than 0.0625 $km^2$ of the terrestrial land of China. The MODIS MOD09Q1 imagery was used to extract surface water (https://ladsweb.modaps.eosdis.nasa.gov/search/). MOD09Q1 is a MODIS level 3 land surface reflectance product. It is an 8-day synthetic imagery of Band 1 (red band) and Band 2 (near-infrared band) with the spatial resolution of 250 m. In this study the near-infrared band is directly used to extract the surface water. There are 22 scenes covering the whole territory of China for every single date in a form of mosaic. For the complete temporal coverage from February 24, 2000 to December 26, 2016, total 16698 images were used. The SRTM (Shuttle Radar Topography Mission) DEM data with 90 m spatial resolution is used as an ancillary data for surface water extraction, which is jointly operated by NASA-JPL (NASA Jet Propulsion Laboratory) and NIMA (National Imagery and Mapping Agency (Slater et al., 2006).

Two types of reference dataset are used for cross comparison. The first is a derived sub-dataset of surface water from China national 30 m land cover dataset of 2000, 2005 and 2010 (Liu et al., 2014; Wu et al. 2017). The second is the global

1    surface water (GSW) at 30-meter resolution from 2000-2015 produced by Pekel et al. (2016).

**3 Methods**

The threshold segmentation method proposed by Lu et al. (2017) which employs single band with one-by-one segmentation

of water bodies is used to extract the surface water boundary, which includes four steps: interferences removal, preliminary

water surface mapping, annual water surface mask acquisition, and water surface boundary extraction (Figure 1). In this

study the last two steps of the method are updated and improved as in following sections 3.1 and 3.2.

<Figure 1>

**3.1 Annual water surface mask acquisition**

The water surface mask is a key input data for excluding land disturbance factors that affect the extraction of the water

surface boundary. It is generated from the preliminary water surface mapping results based on the modified Otsu threshold

method applied on the selected images having less cloud cover and better quality in each year (Lu et al., 2017). In order to

eliminate error in water area information caused by the cloud and cloud shadow in this process, the determination probability

($p$) parameter is used based on the fact that the cloud and its shadow will not appear in the same position for several days.

The equation is as follows,

$$\sum_{i=1}^{n} d_i \geq n \times p,\ D=1$$

where $n$ is the number of the preliminary water surface mapping images, $d_i$ is the pixel value of image $i$, $D$ is the pixel

value of the annual water surface mask, $p$ is the determination probability for identifying water pixel. In this study the

reference images from 2013 to 2016 were selected and the determination probability ($p$) was determined based on the same

rule with Lu et al. (2017). Furthermore, the annual reference images and determination probability ($p$) of 2000-2012 are

directly used here because they were originally obtained based on the whole images of China (Table 2).

<Table 2>

**3.2 Final water surface mapping**

Before determining the threshold value for each water body in the final step of the water surface extraction method (Lu et al.,

2017), the average pixel value in the mask area is used to eliminate the influence of the land pixels. Although this way can

1. improve the accuracy of water surface extraction, the average pixel value in different seasons will also be different. In order

2. to optimize this process, 423 samples of lake and river in different regions of the country are selected (Figure 2) to obtain a

3. reference average pixel value in different seasons. Two images with fewer clouds are selected for each season in each year,

4. and the average pixel values for spring, summer, and autumn are calculated based on the water body samples. They were

5. used as the upper limit threshold for determining the pixel value range for the final step of water surface mapping. In the

6. process of water turning into ice in winter, the pixel value of ice is higher than that of water, and it accounts for a large

7. proportion. The average pixel value will cause the ice layer to be extracted as the water surface, the minimum pixel value of

8. the samples are used as the upper limit threshold for water surface mapping in winter. Finally, based on the upper limit

9. thresholds in different seasons each year, the final binary water surface images of different time period are obtained by using

10. the modified Otsu threshold method again (Lu et al., 2017).

11. <Figure 2>

12. **4 Accuracy assessment**

13. **4.1 Comparison with the national land cover dataset**

14. Based on the 30 m resolution national land cover dataset of 2000, 2005, and 2010, 511 samples from lakes and rivers

15. spreading out across the country are selected as ground truth data (Figure 2), including 11 very large water bodies with areas

16. larger than 1000 km$^2$, 12 large water bodies with areas larger than 500 km$^2$ and less than 1000 km$^2$, 29 medium sized water

17. bodies with area larger than 100 km$^2$ and less than 500 km$^2$, and 459 smaller water bodies with areas less than 100 km$^2$. They

18. were compared with the maximum ISWDC in the corresponding years.

19. The results show that the ISWDC are highly consistent with the reference land cover derived surface water data. The

20. coefficient of determination (R$^2$) in 2000, 2005 and 2010 are found to be 0.9974, 0.992, and 0.9932, respectively as shown in

21. Figure 3. The confusion matrix analysis results show that the average user accuracy is 91.13%, the average producer

22. accuracy is 88.95%, and the average Kappa coefficient is 0.88 in three years (Table 3).

23. As the national land cover data in 2000, 2005, 2010 are based on 30 m Landsat images that mainly obtained in summer

24. season. The water surface in these datasets can be equated with annual maximum water surface results. So we compared

25. them with our maximum ISWDC of corresponding year. The calculated R$^2$ is based on the area of different size of water

bodies. The larger the $R^2$, the better the consistency and the smaller the area error between the two datasets. Furthermore, the

results of confusion matrix are equivalent to pixel scale analysis although it is not as intuitive as visual contrast.

<Figure 3>

<Table 3>

**4.2 Assessment against the global surface water dataset**

The time series of annual ISWDC and GSW permanent water bodies with an area larger than 0.0625 $km^2$ of the whole of

China from 2000-2015 were also compared. The results show that the two datasets possess very good consistency

($R^2$=0.6532) (Figure 4a) and very similar change dynamics (Figure 4b). The annual ISWDC and GSW permanent water

bodies in 2015 also indicate similar spatial patterns in different regions (Figure 5). For the lake groups in central

Qinghai-Tibetan Plateau, the comparison between ISWDC obtained from MODIS and Landsat derived GSW indicated a

closer pattern between the two results (Figure 5a). For the rivers and lakes interlaced with Poyang Lake region, in addition to

the narrow width of the river and some small water bodies, the coincidence between the two datasets is also very high

(Figure 5b). The over-extracted water (red regions in Figure 5) on the margins for large water bodies like Siling Co, Namco,

Poyang Lake, and some of the wide rivers, and the under-extracted slender rivers and small water bodies (green regions in

Figure 5), for the ISWDC dataset, are mainly caused by the mixed pixel effects due to relatively coarse spatial resolution of

the MODIS images.

<Figure 4>

<Figure 5>

**5 Applications and data availability**

**5.1 Time series of surface water dataset applications**

Time series of surface water dataset can be used to analyze the inter-annual and seasonal variation characteristics of surface

water area, including inter-annual variation trend, abrupt change time, intra-annual hydrological process monitoring etc.

(Huang et al., 2018; Xing et al., 2018). Similarly, it can also be used as cross-validation reference data for global surface

water datasets with a similar spatial resolution (Klein et al., 2017), and as a key input parameter for regional and global

hydro-climatic model calibration and evaluation (Khan et al., 2011; Stacke and Hagemann, 2012).

For example, based on the ISWDC from 2000-2016, the annual variation of surface water in China can be obtained by superimposing all the 8-day time series water surface area data of each year. Figure 6 shows that the surface water area began to increase in early March and increased gradually in spring and summer. After reaching its peak in autumn, it then began to decrease gradually. The annual variation of surface water area in different regions can also be portrayed by calculating the multi-year average of every 8-day data. Figure 7 shows that the surface water area of Southwest China (SW) and Northwest China (NW) is very large and inter-seasonally it varies greatly than the surface water area of other regions. Surface water area in Northeast China (NE) began to increase rapidly in spring. It reached a peak in May and decreased slightly in June-July. After reaching its maximum in August-September, it began to decline again in October. In North China (NC), surface water area is relatively small, but the change still shows some seasonality. There is a significant increase in summer and autumn, but the range of increase and decrease is relatively small. Surface water area in Central China (CC) and Eastern China (EC) varies steadily during the year. It reaches its maximum in summer and begins to decrease gradually in late summer and early autumn. Surface water area in South China (SC) was relatively stable throughout the year.

Furthermore, the spatial distributions of surface water can clearly be depicted by means of multi-year average analysis. The results in Table 4 show that surface water of inland China is mainly distributed in western China, accounting for 49.13% of the total surface water area, with 29.88% in the Southwest China (SW) and 19.25% in the Northwest China (NW), followed by the Central China (CC) and East China (EC), which accounted for 8.13% and 24.78% of the total surface water area, respectively. The North China (NC), Northeast China (NE) and South China (SC) account for the other 17.96% of the national surface water area.

<Figure 6>

<Figure 7>

<Table 4>

## 5.2 Data availability

The ISWDC dataset is distributed under a Creative Commons Attribution 4.0 License. The data may be downloaded from the data repository Zenodo at http://doi.org/10.5281/zenodo.2616035 (Lu et al., 2019). In each 8-day surface water image, the pixel values of 1 and 0 represent the water and the background respectively. The 8-day data in each month can be used to

calculate the monthly water occurrence and all the 8-day data in each year can be used to calculate the yearly water occurrence, by summing up all the surface water images together in corresponding time periods. The vector datasets of the 8-day surface water boundaries extracted from the raster data products can also be obtained through the same link.

**6. Discussion and conclusions**

In this study, the 8-day 250-meter resolution surface water dataset of inland China (ISWDC) from 2000 to 2016 has been introduced. It is a fully public sharing data product with prominent features of long time series, moderate spatial resolution and high temporal resolution. The ISWDC is a valuable basic data source for the analysis of dynamic changes of surface water in China in the past 20 years.

The results have been validated based on the 2000, 2005 and 2010 national land cover derived surface water data and show high accuracy. The average user accuracy is 91.13%, the average producer accuracy is 88.95%, and the average Kappa coefficient is 0.88 for these three years. Furthermore, a comparison with the GWS service underlines the reliability of temporal processes and spatial distribution. In terms of temporal variation, the ISWDC and the GWS possess excellent consistency and very similar change dynamics during the whole time period, which simply shows that both datasets are highly correlated. For the spatial distribution characteristics, the ISWDC in 2015 has similar spatial patterns in different regions to that of the GSW dataset, especially for larger water bodies, such as lakes, water reservoirs and wide rivers.

The advantage of the ISWDC dataset is its high level of revealing the spatio-temporal variability of inland surface water. Based on this dataset, the spatial distribution characteristics and temporal variation processes of surface water can be described through the multi-year average spatial statistics and annual data overlapping analysis. In addition, the dataset can also be used as a cross-validation reference data for other global surface water datasets, and a key input parameter for regional and global hydro-climatic models.

However influenced by the algorithm design and the used data sources the results have certain limitations. First of all, as for other surface water datasets derived from multi-spectral sensors, it only includes open water surfaces, while water bodies which are covered by vegetation are not captured. Secondly, as ISWDC only uses MODIS MOD09Q1 near-infrared band for water surface extraction, thus the accuracy of datasets depends mainly on the quality of the original 8-day synthetic images.

When there exist clouds exist in the water distribution region of the synthetic image at a certain time, the cloud covered water surface will not be extracted which causes underestimation for extracting water bodies. In addition, the reference images used to produce the annual water surface mask will also affect the accuracy of the final results. For example, if the selected image does not contain the information of the actual maximum water surface occurrence in that year, it may lead to the exclusion of that part of the water pixels which lies outside the mask. Finally, because of the small difference of reflectance between the ice-water mixing boundary in autumn and spring, the accuracy of water surface area extraction will be limited in these two seasons.

Although the water surface extraction method designed in this study is aimed at extracting water surface information from the MODIS MOD09Q1 images, its core process is automatic thresholding for estimation of water bodies one by one. Therefore, this method is also applicable to traditional water body indices, such as NDWI, MNDWI and AWEI, or to other water surface information based on enhanced thematic data. In the future, while continuing to extend the existing datasets from 2017 to now by using this method, the 30-meter GWS dataset in China will be extended. At the same time, the national 10-meter spatial resolution water surface dataset based on Sentinel-2 imagery will be produced. After the national scale datasets are completed, the corresponding global scale datasets are also expected.

**Author contributions.** SL supervised the downloading and processing of satellite images and designed the methodology. JM contributed to downloading, processing satellite images, and extracting the surface water data (ISWDC). XM extracted the reference surface water data from the national land cover datasets and analyzed the accuracy of the ISWDC. HT extracted the Global Surface Water (GSW) from the Google Earth Engine platform. HL made contribution for manuscript structure design and revision. MHAB optimized article structure, figures and English grammar. All authors have read and approved the final paper.

**Competing interests.** The authors declare that they have no conflict of interest.

**Acknowledgements.** We thank the National Key Research and Development Program of China (2017YFC0405802,

2016YFC0503507-03), the Key Program of the National Natural Science Foundation of China (91637209), the project of China geological survey (DD20160106), and the Strategic Priority Research Program of the Chinese Academy of Sciences (XDA19070201) for financial support. We thank NASA EOSDIS LAADS DAAC platform (https://ladsweb.modaps.eosdis.nasa.gov/) and NASA-JPL and NIMA for providing the MODIS and SRTM datasets. We also thank JRC and Google Earth Engine (https://earthengine.google.com) for providing the Global Surface Water (GSW) dataset.

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

Table 1. National and regional surface water related datasets of China.

| Dataset | Author | Time series | Resolution |
|---------|--------|-------------|------------|
| Lake water surface of Tibetan Plateau | Lu et al., 2017 | 8-days, 2000-2012 | 250m |
| Lake surface area of Tibetan Plateau | Song et al., 2013 | 1970s, 1990, 2000, 2003-2009, 2011 | 60m, 30m |
| Lake area of Tibetan Plateau | Zhang et al., 2014, 2017 | 1970s, 1990, 2000, 2010 | 15m, 30m |
| A lake dataset for the Tibetan Plateau | Wan et al., 2014, 2016 | 1960s, 2005, 2014 | 16m, 30m |
| China national wetland datasets | Niu et al., 2012 | 1978, 1990, 2000, 2008 | 30 m |
| China national land cover datasets | Wu et al., 2017 | 1990, 2000, 2010, and 2015 | 30 m |
| China national land use datasets | Liu et al., 2018 | 1990, 1995, 2000, 2005, 2010, 2015 | 30m |

Table 2. The images used for annual water surface mask generation and the determination probability each year.

| Year | Selected 8-day image dates (DOY) | Determination probability ($p$) |
|------|----------------------------------|--------------------------------|
| 2000 | 185、201、209、233、241、249、257、265、281、305 | 0.2 |
| 2001 | 185、193、201、233、241、249、257、265、273、281 | 0.2 |
| 2002 | 185、193、209、217、225、233、241、249、257、265 | 0.2 |
| 2003 | 177、193、201、209、217、233、249、257、265、289 | 0.3 |
| 2004 | 185、201、217、225、233、249、257、265、273、281 | 0.2 |
| 2005 | 209、217、225、233、241、249、257、265、273、281 | 0.2 |
| 2006 | 137、145、169、177、185、193、201、209 | 0.2 |
| 2007 | 185、193、201、209、217、225、233、241、257、265 | 0.3 |
| 2008 | 193、201、209、225、233、241、249、257、265、273 | 0.3 |
| 2009 | 129、137、153、169、185、193、201、233、241、249 | 0.3 |
| 2010 | 185、209、217、225、233、241、249、257、273、281 | 0.2 |
| 2011 | 161、169、177、185、201、209、217、225、233、265 | 0.2 |

| 2012 | 185、201、209、217、225、233、241、257、265、273 | 0.2 |
| 2013 | 185、193、201、209、217、225、233、249、257、281 | 0.2 |
| 2014 | 193、201、209、225、233、241、249、265、257 、273 | 0.3 |
| 2015 | 201、209、217、241、249、257、265、273、281、289 | 0.2 |
| 2016 | 193、209、225、241、257、265、273、289、305 | 0.2 |

2 **Table 3. Accuracy analysis samples in different region and the accuracy evaluation results.**

| Sample regions | Sample water bodies | | | | |
| --- | --- | --- | --- | --- | --- |
| | Very large | Large | Medium | Small | Total |
| North China (NC) | 1 | 1 | 1 | 73 | 76 |
| Northeast China (NE) | 1 | 2 | 2 | 21 | 26 |
| East China (EC) | 2 | 1 | 3 | 34 | 40 |
| Southwest China (SW) | 2 | 3 | 5 | 75 | 85 |
| Northwest China (NW) | 2 | 2 | 13 | 166 | 183 |
| Central China (CC) | 2 | 1 | 2 | 46 | 51 |
| South China (SC) | 1 | 2 | 3 | 44 | 50 |
| Average user accuracy | 96.14 | 94.75 | 93.69 | 79.96 | 91.13 |
| Average producer accuracy | 92.64 | 88.87 | 92.69 | 81.60 | 88.95 |
| Average Kappa coefficient | 0.94 | 0.93 | 0.93 | 0.72 | 0.88 |

4 **Table 4. The average distribution of surface water area in inland China from 2000-2016**

| Regions | Area($km^2$) | Area percentage (%) |
| --- | --- | --- |
| North China (NC) | 6250.6 | 6.11 |
| Northeast China (NE) | 8991.3 | 8.79 |

| | | |
|---|---|---|
| East China (EC) | 25342.3 | 24.78 |
| Central China (CC) | 9313.4 | 8.13 |
| South China (SC) | 3126.0 | 3.06 |
| Southwest China (SW) | 30548.6 | 29.88 |
| Northwest China (NW) | 19680.2 | 19.25 |
| Total | 103252.3 | 100.00 |

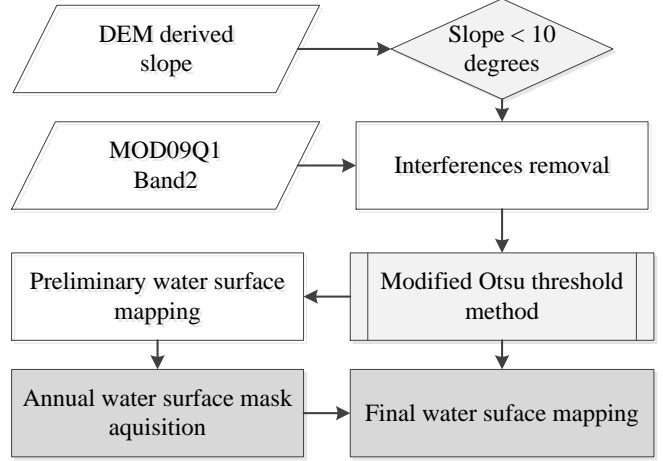

3 **Figure 1. Flowchart of the water surface extraction method reference to Lu et al. (2017).**

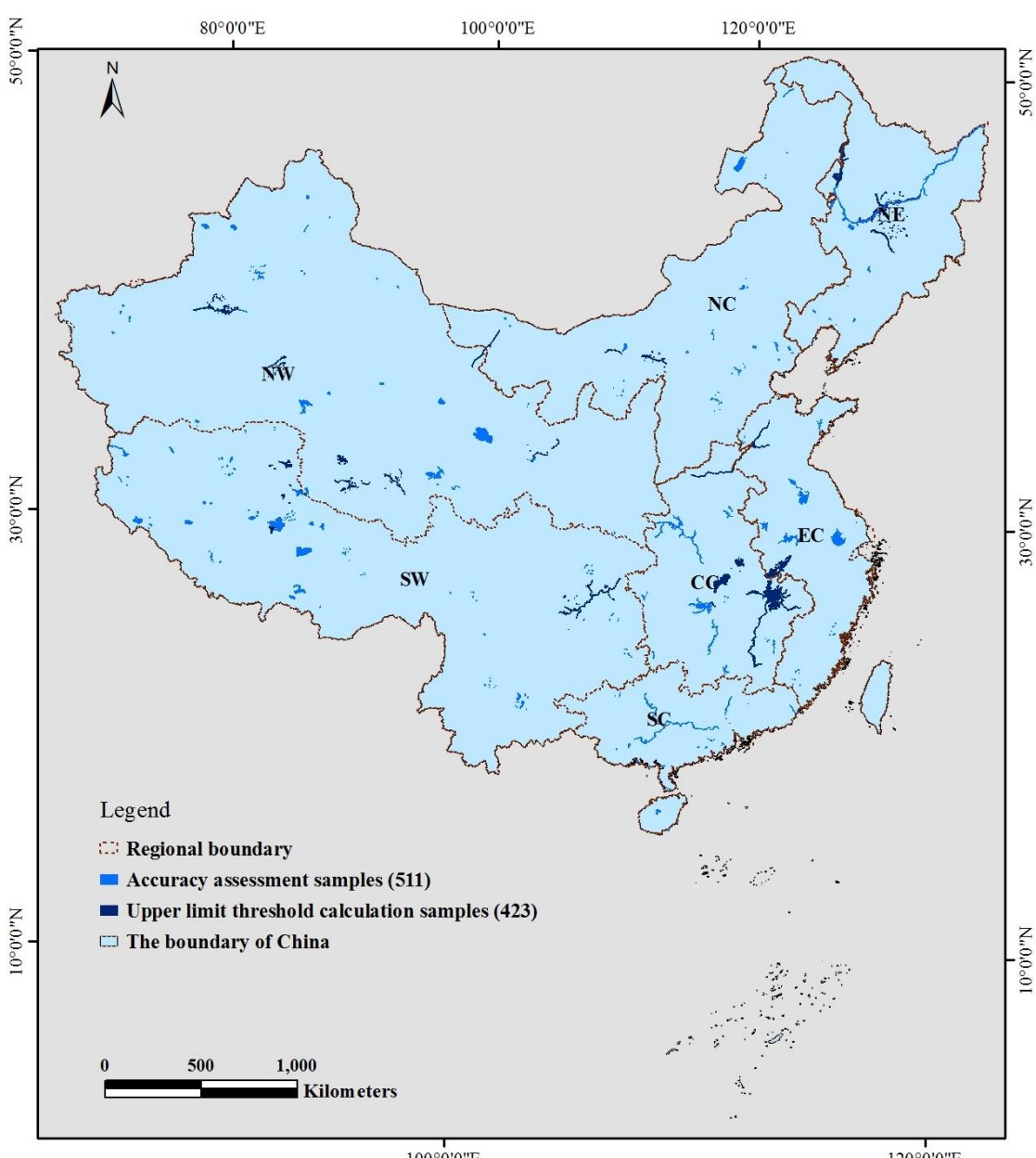

2    **Figure 2. The boundary of China, the accuracy assessment and the upper limit threshold calculation samples for surface water**

3    **extraction. NW: Northwest China, SW: Southwest China, SC: South China, CC: Central China; NC: North China, NE: Northeast**

4    **China, EC: East China.**

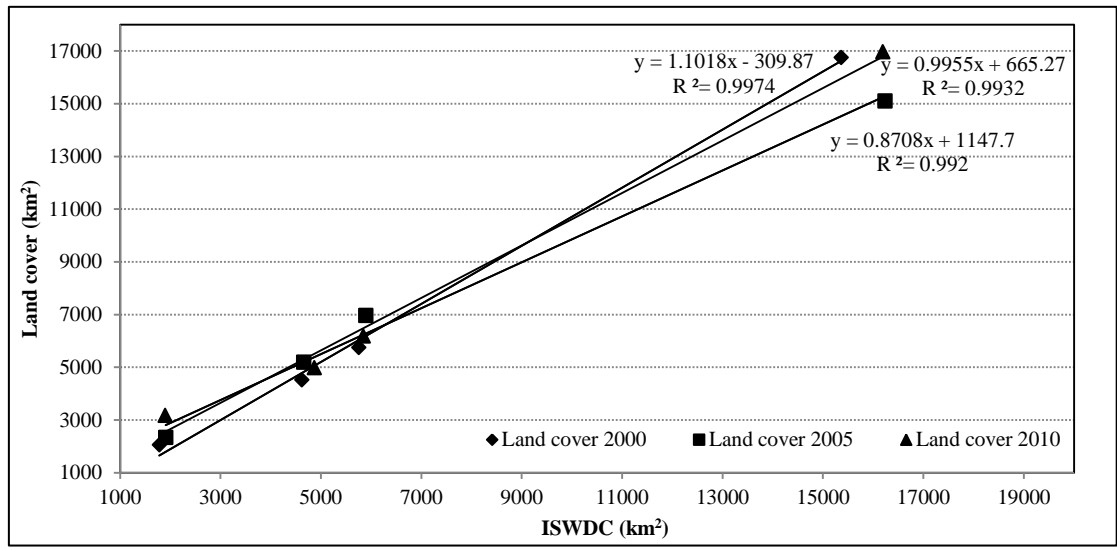

2 **Figure 3. Comparison of the total area of surface water body samples with different size (< 100 km2, 100-500 km2, 500-1000**

3 **km2, >1000 km2) between ISWDC and the National land cover derived surface water data.**

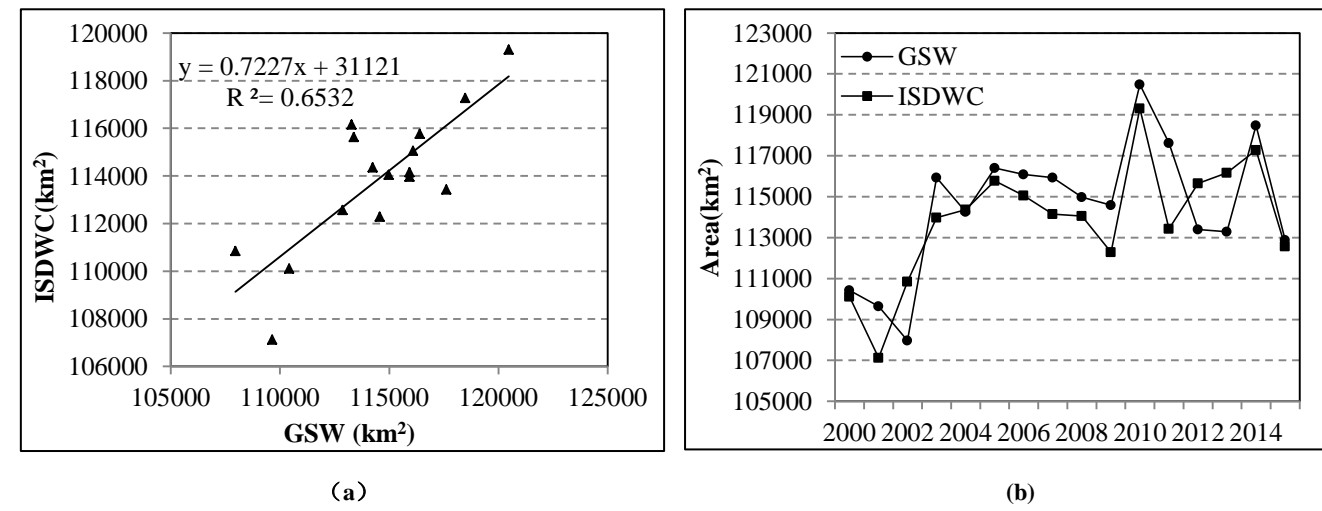

6 (a)                                            (b)

7 **Figure 4. Comparison of the time series annual ISWDC and GSW permanent water bodies of the whole of China from 2000-2015.**

8 **(a) is the correlation analysis result, (b) is the change trend comparison result.**

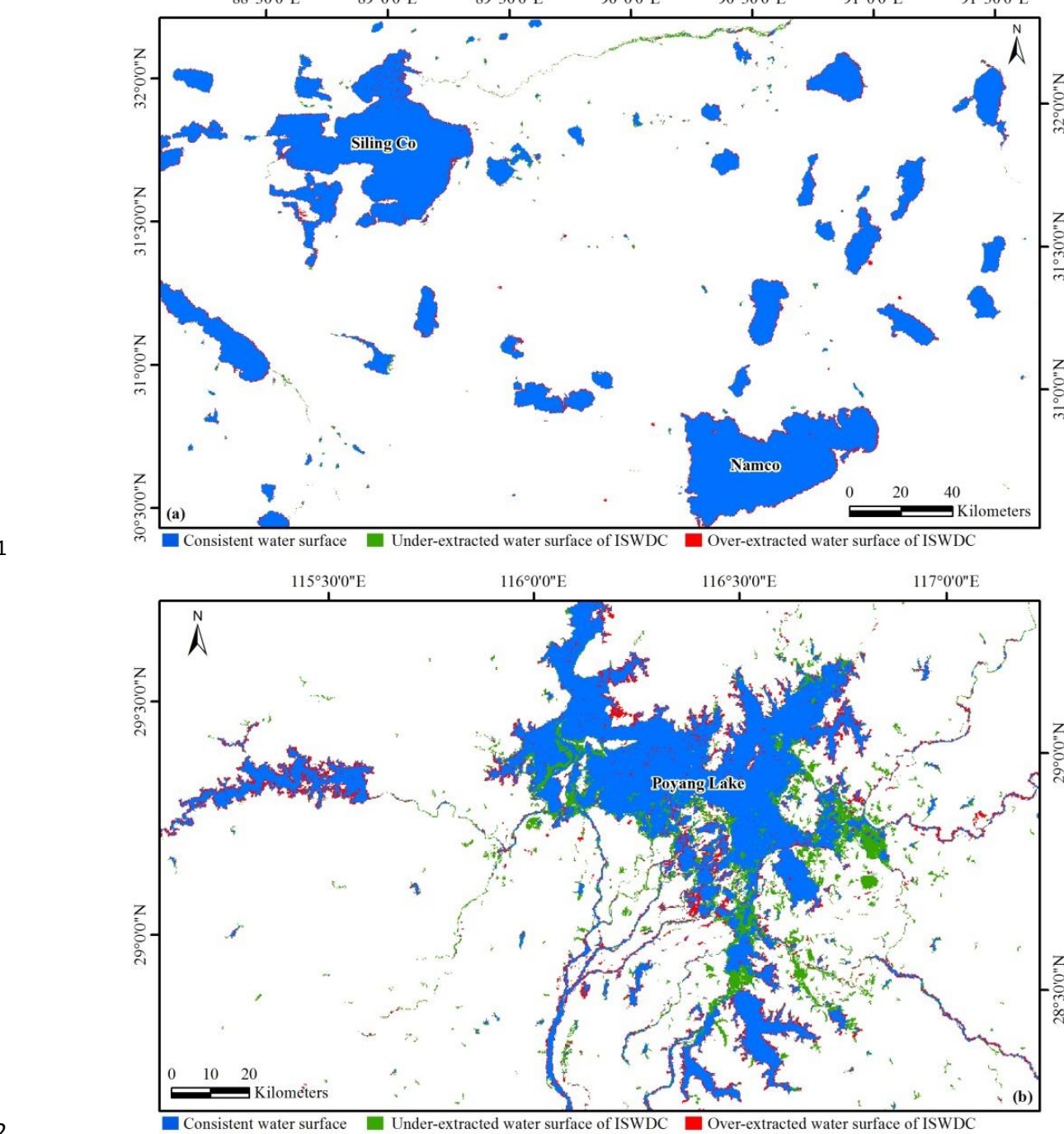

**Figure 5. Comparison of permanent water bodies derived from ISWDC and GSW over the sites of the central Qinghai-Tibetan Plateau (a) and Poyang Lake region (b).**

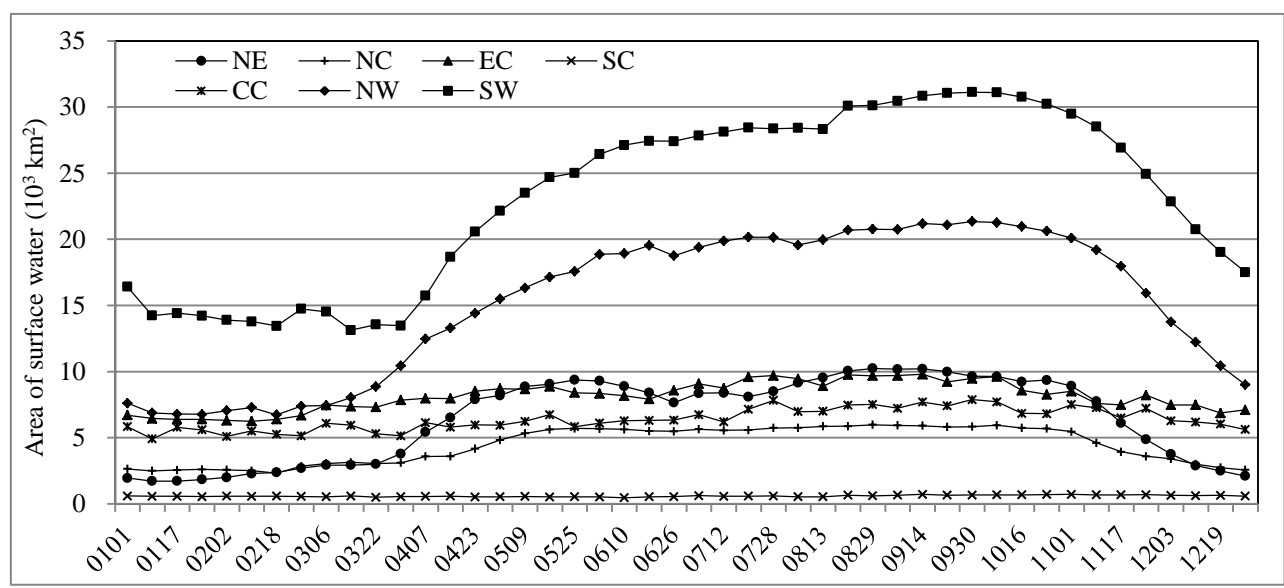

2    **Figure 6. Annual change of total water area during the period of 2000-2016.**

4    **Figure 7. The 8-day surface water area in different regions of China from 2000 to 2016. NE: Northeast China, NC: North China,**

5    **EC: East China, SC: South China, CC: Central China, NW: Northwest China, SW: Southwest China.**