# Peer review of "Time series of Inland Surface Water Dataset in China (ISWDC) for 2000-2016"

_Earth System Science Data, 2018_

## Referee Comment (RC1) · Anonymous Referee #1 · 18 Jan 2019

Manuscript:

This manuscript showed some potential for publication. However, after looking into more detail it shows (1) a lot of duplication (previous paper Lu et a., 2017), (2) lack of sufficient accuracy assessment, (3) some incorrect quotes and very general statements. In terms of methodology this manuscript is almost exactly the same as presented by Lu et al., 2017. I do not disagree with the idea of using same methodology for a larger area of interest. But the methodological paragraphs are 90% identically with even same Figure and Table. Some adjustments are mentioned but are not discussed. In terms of accuracy assessment there is a lack of detailed analyses of the presented

product. The presented numbers are based on calculated total areas over the entire region for maximum water surface area and do not even analyze the temporal scale. The second dataset for cross comparison GWS (Pekel et al., 2016), was only used to compare visually for two zoom-in areas. With the final statement that both datasets look alike and have similar patterns. The temporal resolution is not validated at all. The effort of accuracy assessment which was performed is not state of the art and does not mirror the real quality of the presented product. In my view this manuscript does not have much of novelty and the statement about "lack of water surface products for China" is also not correct.

General Comments:

I would recommend English gramma check. Some sentences are understandable but have incorrect gramma (e.g. P1 L16, P1 L21, P 2 L5-6, P2 L17, P4 L2, P4 L10-11 and more) Please be consistent: either dataset or data set. Sometimes you use both options in one paragraph (e.g. P4) The methodological content is almost identical in many parts with Lu et al., 2017

Abstract and conclusion: Almost the same content!

Introduction

P 2 L 2: First statement -> please provide a source

P2 L 15: statement incorrect. The data is being produced from July 2002-ongoing.

P2 L19-24: this information could be summarized with key parameters in a table

P3 L3-4: NDWI is mostly known from McFeeters 1996 as well as Gao 1996

P3 L5: wrong citation of Feyisa et al. 2014 instead of 2018

P3 L7-8: incorrect statement for Pekel et al., 2014

P3 L24: LBV?

P4 L2: first sentence does not make sense.

P4 L9-11: This statement is not true as there are datasets which reflects the spatial and temporal characteristics of surface water such as Pekel et al., 2016, Klein et al., 2017, Ji et al., 2018.

P4 L14: why does ISWDC ends with 2016?

P5 L16: Something should be following. Instead the paragraph ends here.

P5 Figure 1 is exactly the same as in Lu et al., 2017 if you have improved two steps, why don't update the figure as well and discuss the adjustments?

P5-7: is almost identically with your paper about Tibetan Plateau (TB). Even the table 1 with selected images is the same. Was there no difference of analyzing only TB and entire China? Maybe instead of copy-pasting the same text, you should clearly point out the updates and improvement which you did. At the moment, you only mention there were improvements but you do not mention where exactly these improvements are and why it was necessary.

Paragraph 3.2. boundary extraction? Your dataset is a raster data with pixel values for water and no water. For water boundary I would expect a vector dataset with lines or polygons which determines the boundaries of a water body. This paragraph is unclear and does not correspond to the header of the paragraph.

P7 L10-13: why is the difference between ice layer and water body in winter small?

4 Accuracy Assessment

P9 L7-8: what is the logic behind this procedure? You are comparing a static dataset of a certain year with the maximum area of ISWDC of corresponding year. What is the interpretation of calculated $R^2$ in that case? To compare the total area over such a large area of all selected water bodies seems to be very shallow and not state of the art. I would expect a pixel based approach to actually assess the real quality at a

certain time for a certain pixel.

Paragraph 4.2: it is only a visual interpretation of two zoom-in images without any quantitative results.

5.1 The multi-year average analysis can also be done with GSW dataset. Therefore, I don't see any novelty or interesting facts especially since you mentioned that both datasets indicate similar patterns. Where is the advantage or novelty of using ISDWC in this case?

P13 Figure 5: this figure shows that, in general, the total surface water of China has a clear seasonality. However, the curves of single years cannot be determined due to figure design.

P13 Figure 6: In this Figure you show red lines, which are probably the water body borders. Are these polygons also distributed? So far I was only able to download binary raster data and no vector data.

P14 L11-12: again a statement which I do not agree with as the presented dataset covers almost the same time span as it is covered by GWS (Pekel et al., 2016).

Conclusion: is kept very general.

Dataset

The dataset itself is a mosaic for China area for temporal steps of 8 days and can used by interested scientists or organizations for different purposes. However, without sufficient quality layer or accuracy assessment especially of the different time steps.

---

## Referee Comment (RC2) · Anonymous Referee #2 · 2 Feb 2019

General Remarks The paper is in general well written but lacks the bigger picture. The work reported understandably China focused but the authors miss the opportunity to discuss how their techniques could be applied elsewhere and what the long term benefits are. It is highly recommended that such a discussion be included.

Although a degree of statistical testing has been applied, the results of which are reported, the paper lacks a discussion of the overall uncertainty associated with the surface water areas reported. It is highly recommended that such a discussion be added.

Section 5.2 Although the links to the data work and there is a 'ReadMe' file accompa-

nying the data the metadata it contains is minimal. This section needs expansion to include a description of the data archive structure, access, and usage licensing as well as the file formats and metadata provided.

The authors are providing imagery (including .tif format). It is advised that the metadata in the 'ReadMe' file be embedded into the files - this would improve the usability of the data in the long-term. For example the following could be used and or adapted to meet the authors needs https://iptc.org/standards/photo-metadata/photo-metadata/

Specific edits Page 1 15: 'for the time' change to 'for time' 16: 'create Inland' change to 'create an Inland' 16: 'maps the water body' change to 'maps water bodies' 17: '0.0625 km2 17 in the terrestrial land of China for the period 2000–2016, in 8-day temporal' change to '0.0625 km2 17 within the land mass of China for the period 2000–2016, with 8-day temporal' 18: remove 'the' at end of line 20: 'data with the' change to 'data with' 21: '2015 too' change to '2015' 23: 'and as input' change to 'and as an input'

Page 2 3: 'systems in' change to 'systems in a' 5: 'has a role' change to 'have a role' 9: 'But' change to 'but' 9: 'did limited exploration for' change to 'were limited' 17: 'but in' change to 'but only in' 17 – 19: 'Their research hotspot was Qinghai-Tibetan Plateau due to the existence of the largest number of inland lakes there with the highest elevation on the planet (Lu et al., 2017).' This sentence does not make any sense and needs restructuring. 20: 'Almost every 10-year of lake water surface area datasets from 1960s to present has been produced' change to 'Almost every 10-year since the 1960s lake water surface area datasets have been produced'

Page 3 1: 'dataset.' change to 'datasets are available.' 3: 'is water' change to 'is a water' 3: 'as Normalized' change to ' as the Normalized' 8: 'these methods to extract water boundary is to determine' change to 'these methods in extracting the water boundary is to determine' 11: 'experience causes' change to 'experience which causes' 11: 'and it is' change to 'and is' 12: 'apply to large scale and large amount of data research' change to 'apply on larger scales and to large amounts of data' 16: 'and divided it'

change to 'and to divide these' 18: 'of visual' change to 'of a visual'

Page 4 2: 'China is one of the most rivers and lakes in the world' change to 'China has one of the highest densities of rivers and lakes in the world' 3: 'exceeding 1000 km2, and 2928 lakes with an area larger than 1 km2 and a total area of 91,020 km2 (Ma' change to 'exceeding 1000 km2, 2928 lakes with an area larger than 1 km2 giving in total a surface water area of 91,020 km2 (Ma' 5: 'resources are very uneven in distribution.' Change to 'resources are unevenly distributed.' 7: 'bought' change to 'placed' 10: 'China. So the research to' change to 'China, hence the potential to' 12: Remove 'Therefore' and 'research' 17: Replace 'other' with 'existing' 20: 'to the water' change to 'to a water'

Page 5 4: 'as an ancillary' change to 'as ancillary' 7: 'The first one' change to 'The first' 9: 'The second one' change to 'The second'

Page 6 4: 'extraction of' change to 'extraction of the' 7. 'the cloud and cloud shadows in this process' change to 'cloud and cloud shadow in this process' 8: What is the time period referred to by the statement 'over longer time periods' 10: remove 'if ' before equation

Page 7 5: 'will be' change to 'will also be' 8: 'values of' change to 'values for' 10: 'values of' change to 'values for' 12: 'extracted as' change to 'extracted as the'

Page 9 9: 'high consistency' change to 'highly consistent' 11: 'respectively (Figure 3).' change to 'respectively is shown in figure 3.'

Page 12 3: 'series surface' change to 'series of the surface' 4: 'area such as' change to 'area; including; 6: 'as a cross-validation' remove 'a' 8: 'models' change to 'model' Table 19: need to include uncertainty accessioned with the values given

Page 13 Figure 5: There are no error bars on this figure – they need to be added or an explanation as to why they are not shown. The axis tick marks need to be 'out' rather than 'in' to improve clarity. The x axis labelling is cluttered and needs revision to make

clearer

Page 14 11: '2016 was' change to '2016 has been' 11: 'series and' change to 'series with' 12: 'of surface' change to 'of a surface' 12: 'in China' change to 'for China' 13: 'in high consistency' change to 'is highly consistent' 14: 'data in' change to 'data from' 16: '0.88 in' change to '0.88 for the' 18: 'and Poyang Lake region) with the GSW data set, especially for the large water bodies (as lakes and' change to 'and Poyang Lake region) to that of the GSW data set, especially for large water bodies (such as lakes and' 19: 'and the' change to 'and' 20: remove 'for' 21: 'process' change to 'processes'

---

## Author Comment (AC1) · 29 Mar 2019

Dear Reviewer,

Thank you very much for your constructive and detailed comments which are really very helpful to improve and clarify the manuscript. We have through them in detail and made amends as requested. We provided a very detailed response to each of your comments, and revised the manuscript based on all the comments from other reviewers. We also provided a revised manuscript in tracked changes mode.

Best wishes,

Shanlong

Please also note the supplement to this comment:
https://www.earth-syst-sci-data-discuss.net/essd-2018-134/essd-2018-134-AC1-supplement.zip

---

## Author Comment (AC2) · 29 Mar 2019

Dear Reviewer,

Thank you very much for your comments and suggestions which are very helpful to improve the manuscript. We provided a very detailed response to each of your comments, and revised the manuscript based on all the comments from you and another reviewer. We also provided a revised manuscript in tracked changes mode.

Best wishes,

Shanlong

[Figure]

Please also note the supplement to this comment:
https://www.earth-syst-sci-data-discuss.net/essd-2018-134/essd-2018-134-AC2-supplement.zip
* * *

---

## Author Response (AR1)

**Part 1 Response to the Reviews**

**1. Response to reviewer 1**

**Questions:** This manuscript showed some potential for publication. However, after looking into more detail it shows (1) a lot of duplication (previous paper Lu et a., 2017), (2) lack of sufficient accuracy assessment, (3) some incorrect quotes and very general statements. In terms of methodology this manuscript is almost exactly the same as presented by Lu et al., 2017. I do not disagree with the idea of using same methodology for a larger area of interest. But the methodological paragraphs are 90% identically with even same Figure and Table. Some adjustments are mentioned but are not discussed. In terms of accuracy assessment there is a lack of detailed analyses of the presented product. The presented numbers are based on calculated total areas over the entire region for maximum water surface area and do not even analyze the temporal scale. The second dataset for cross comparison GWS (Pekel et al., 2016), was only used to compare visually for two zoom-in areas. With the final statement that both datasets look alike and have similar patterns. The temporal resolution is not validated at all. The effort of accuracy assessment which was performed is not state of the art and does not mirror the real quality of the presented product. In my view this manuscript does not have much of novelty and the statement about "lack of water surface products for China" is also not correct.

**Responses:** Many thanks to the reviewer for his/her review and comprehensive, in-depth and constructive suggestions. Please find the detailed responses to all the suggestions along with proposed changes below. We also uploaded the updated manuscript using track change (in response to both reviewers) in a separate post.

As for the duplication problem of the method introduction, due to the emphasis of this paper is introduce of the new dataset based results, so in the methodology part we only introduced the supplemented and optimized content (Section 3.1 and 3.2) on the basis of previous achievements.

On the issue of accuracy evaluation, we have increased the accuracy analysis of temporal process data.

We also revised the grammar and inaccurate expressions one by one according to the comments of the reviewers.

**Questions:** General Comments:

I would recommend English gramma check. Some sentences are understandable but have incorrect gramma (e.g. P1 L16, P1 L21, P 2 L5-6, P2 L17, P4 L2, P4 L10-11 and more) Please be consistent: either dataset or data set. Sometimes you use both options in one paragraph (e.g. P4) The methodological content is almost identical in many parts with Lu et al., 2017

Abstract and conclusion: Almost the same content!

**Response:** We have corrected the grammatical and expression errors one by one, and made the abstract and conclusions different from each other.

Introduction

**Question:** P 2 L 2: First statement -> please provide a source

**Response:** We added the reference from Lu and He, 2006.

**Question:** P2 L 15: statement incorrect. The data is being produced from July 2002-ongoing.

**Response:** We revised it as 'However, the temporal resolution of the former research is near monthly, and the latter research only produced datasets of 2013-2015 at the moment, the entire MODIS archive back to July 2002 is still ongoing (Klein et al., 2017).'

**Question:** P2 L19-24: this information could be summarized with key parameters in a table

**Response:** We summarized them into a table named 'Table 1 National and regional surface water related datasets of China'

**Question:** P3 L3-4: NDWI is mostly known from McFeeters 1996 as well as Gao 1996

**Response:** We added these two original references.

**Question:** P3 L5: wrong citation of Feyisa et al. 2014 instead of 2018

**Response:** It is a wrong typing caused problem. We have revised in the new version.

**Question:** P3 L7-8: incorrect statement for Pekel et al., 2014

**Response:** We changed the statement as 'the multiband transformation method (Pekel et al. 2014)'

**Question:** P3 L24: LBV?

**Response:** It is a new transformation method proposed by Zhang et al. (2017). It means: L, the general radiance level; B, the visible–infrared radiation balance; V, the radiance variation vector between bands. We added this explanation in the manuscript.

**Question:** P4 L2: first sentence does not make sense.

**Response:** We revised this sentence as 'China is one of the countries that have the highest densities of rivers and lakes in the world…..'

**Question:** P4 L9-11: This statement is not true as there are datasets which reflects the spatial and temporal characteristics of surface water such as Pekel et al., 2016, Klein et al., 2017, Ji et al., 2018.

**Response:** We changed this statement as 'Therefore, there is an urgent need for spatio-temporal continuous surface water datasets to support the robust and efficient management of water resources, and to investigate the relationship between the national surface water and the global climate and human activities.  However, until now, full public sharing data products with moderate spatial resolution and near-daily temporal resolution are still lacking in China.

**Question:** P4 L14: why does ISWDC ends with 2016?

**Response:** We are going to extend the ISWDC to 2019, when it is finished we will update it onto the zenodo platform.

**Question:** P5 L16: Something should be following. Instead the paragraph ends here.

**Response:** Maybe here is a grammar caused mistake. We did not end the sentence here. The Section 3.1 and 3.2 are explanations of the sentence of 'In this study the last two steps related to 'annual water surface mask acquisition' and 'final water surface mapping' are updated and improved as in following sections 3.1 and 3.2.

**Question:** P5 Figure 1 is exactly the same as in Lu et al., 2017 if you have improved two steps, why don't update the figure as well and discuss the adjustments?

**Response:** Figure 1 shows the core steps of our mapping method for MODIS MOD09Q1 images. As we mentioned in the manuscript, we only updated (Section 3.1) and improved (Section 3.2) the last two steps of the method, the overall structure of the method did not change, so we did not change the flowchart. But we explained the adjustments in Section 3.1 and Section 3.2, and we changed the title of the final step as 'Final water surface mapping'.

**Question:** P5-7: is almost identically with your paper about Tibetan Plateau (TB). Even the table 1 with selected images is the same. Was there no difference of analyzing only TB and entire China? Maybe instead of copy-pasting the same text, you should clearly point out the updates and improvement which you did. At the moment, you only mention there were improvements but you do not mention where exactly these improvements are and why it was necessary.

**Response:** In this new version, we have deleted some duplicated words, and explained the changes made in this study and the contents inherited from Lu et al. (2017).

**Question:** Paragraph 3.2. boundary extraction? Your dataset is a raster data with pixel values for water and no water. For water boundary I would expect a vector dataset with lines or polygons which determines the boundaries of a water body. This paragraph is unclear and does not correspond to the header of the paragraph.

**Response:** The major content of this section is about the improvement of the threshold value determination during the final step of water surface mapping. We changed the title of this section and optimized the description and interpretation.

**Question:** P7 L10-13: why is the difference between ice layer and water body in winter small?

**Response:** Thanks for pointing out this issue. We have corrected this point. In the new version, we changed it as 'In the process of water turning into ice in winter, the pixel value of ice is higher than that of water, and it accounts for a large proportion. The average pixel value will cause the ice layer to be extracted as the water surface.'

4 Accuracy Assessment

**Questions:** P9 L7-8: what is the logic behind this procedure? You are comparing a static dataset of a certain year with the maximum area of ISWDC of corresponding year. What is the interpretation of calculated R2 in that case? To compare the total area over such a large area of all selected water bodies seems to be very shallow and not state of the art. I would expect a pixel based approach to actually assess the real quality at acertain time for a certain pixel.

**Response:** This comparison is very similar as we extracted lot of samples from water bodies from images having 30m resolution and took them as ground truth data for accuracy analysis. As the national land cover data in 2000, 2005, 2010 are based on 30 m Landsat images that mainly obtained in summer season. The water surface in these datasets can be equated with annual maximum water surface results. So we compared them with our maximum ISWDC of corresponding year. The calculated $R^2$ is based on the area of different size of water bodies. The larger the $R^2$, the better the consistency and the smaller the area error between the two datasets. Furthermore, the

results of confusion matrix are equivalent to pixel scale analysis although it's not as intuitive as visual contrast.

**Question:** Paragraph 4.2: it is only a visual interpretation of two zoom-in images without any quantitative results.

**Response:** We added a comparison between the annual permanent water of ISWDC and GSW in whole China in 2000-2015. The results show that the two datasets are also very consistent in time series analysis.

**Question:** 5.1 The multi-year average analysis can also be done with GSW dataset. Therefore, I don't see any novelty or interesting facts especially since you mentioned that both datasets indicate similar patterns. Where is the advantage or novelty of using ISWDC in this case?

**Response:** In this section, we added the results of time series analysis of surface water change in China and different geographical regions with the ISWDC. These results fully reflect the characteristics and advantages of high temporal resolution of the ISWDC.

**Question:** P13 Figure 5: this figure shows that, in general, the total surface water of China has a clear seasonality. However, the curves of single years cannot be determined due to figure design.

**Response:** We are trying to use the 8-day time series surface water area data from 2000 to 2016 to show the inter-annual and annual changes (see the following figure), but with this figure the seasonality characteristic is obscured. So the original figure is retained but we optimized the figure.

[Figure]

**Question:** P13 Figure 6: In this Figure you show red lines, which are probably the water body borders. Are these polygons also distributed? So far I was only able to download binary raster data and no vector data.

**Response:** Yes, the red lines in the original Figure 6 are the water body borders that we extracted with the binary raster data. Although we delete this figure in our new version, we will distribute the vector version of ISWDC while submitting revised article.

**Question:** P14 L11-12: again a statement which I do not agree with as the presented dataset covers almost the same time span as it is covered by GWS (Pekel et al., 2016).

**Response:** We deleted this statement and changed it to 'It is a full public sharing long time series data product with moderate spatial resolution and high temporal resolution, and is a very good basic data source for the analysis of the dynamic changes of surface water in China and regions in the past 20 years'.

**Question:** The dataset itself is a mosaic for China area for temporal steps of 8 days and can used by interested scientists or organizations for different purposes. However, without sufficient quality layer or accuracy assessment especially of the different time steps.

**Response:** In the new version of the dataset, the vector files are added, and accuracy evaluation and possible problems are explained.

**2. Response to reviewer 2**

General Remarks

**Questions:** The paper is in general well written but lacks the bigger picture. The work reported understandably China focused but the authors miss the opportunity to discuss how their techniques could be applied elsewhere and what the long term benefits are. It is highly recommended that such a discussion be included.

**Response:** In the new version of manuscript, we changed the title of Section 6 to Discussion and conclusions, and extended the final paragraph to discuss the application of our method, datasets and future plan.

**Questions:** Although a degree of statistical testing has been applied, the results of which are reported, the paper lacks a discussion of the overall uncertainty associated with the surface water areas reported. It is highly recommended that such a discussion be added.

**Response:** In the new final section, we added a paragraph to discuss the overall uncertainties.

**Questions:** Section 5.2 Although the links to the data work and there is a 'ReadMe' file accompanying the data the metadata it contains is minimal. This section needs expansion to include a description of the data archive structure, access, and usage licensing as well as the file formats and metadata provided.

The authors are providing imagery (including .tif format). It is advised that the metadata in the 'ReadMe' file be embedded into the files - this would improve the usability of the data in the long-term. For example the following could be used and or adapted to meet the authors needs https://iptc.org/standards/photo-metadata/photo-metadata/

**Response:** We have added vector data of the datasets and detail introduction of the data using and transferring.

Specific edits

Page 1 15: 'for the time' change to 'for time' 16: 'create Inland' change to 'create an Inland' 16: 'maps the water body' change to 'maps water bodies' 17: '0.0625 km2 17 in the terrestrial land of China for the period 2000–2016, in 8-day temporal' change to '0.0625 km2 17 within the land mass of China for the period 2000–2016, with 8-day temporal' 18: remove 'the' at end of line 20: 'data with the' change to 'data with' 21: '2015 too' change to '2015' 23: 'and as input' change to 'and as an input'.

Page 2 3: 'systems in' change to 'systems in a' 5: 'has a role' change to 'have a role' 9: 'But' change to 'but' 9: 'did limited exploration for' change to 'were limited' 17: 'but in' change to 'but only in' 17 – 19: 'Their research hotspot was Qinghai-Tibetan Plateau due to the existence of the largest number of inland lakes there with the highest elevation on the planet (Lu et al., 2017).' This sentence does not make any sense and needs restructuring. 20: 'Almost every 10-year of lake water surface area datasets from 1960s to present has been produced' change to 'Almost every 10-year since the 1960s lake water surface area datasets have been produced'

Page 3 1: 'dataset.' change to 'datasets are available.' 3: 'is water' change to 'is a water' 3: 'as Normalized' change to ' as the Normalized' 8: 'these methods to extract water boundary is to determine' change to 'these methods in extracting the water boundary is to determine' 11: 'experience causes' change to 'experience which causes' 11: 'and it is' change to 'and is' 12: 'apply to large scale and large amount of data research' change to 'apply on larger scales and to large amounts of data' 16: 'and divided it' change to 'and to divide these' 18: 'of visual' change to 'of a visual'

Page 4 2: 'China is one of the most rivers and lakes in the world' change to 'China has one of the highest densities of rivers and lakes in the world' 3: 'exceeding 1000 km2, and 2928 lakes with an area larger than 1 km2 and a total area of 91,020 km2 (Ma' change to 'exceeding 1000 km2, 2928 lakes with an area larger than 1 km2 giving in total a surface water area of 91,020 km2 (Ma' 5: 'resources are very uneven in distribution.' Change to 'resources are unevenly distributed.' 7: 'bought' change to

'placed' 10: 'China. So the research to' change to 'China, hence the potential to' 12: Remove 'Therefore' and 'research' 17: Replace 'other' with 'existing' 20: 'to the water' change to 'to a water'

Page 5 4: 'as an ancillary' change to 'as ancillary' 7: 'The first one' change to 'The first' 9: 'The second one' change to 'The second'

Page 6 4: 'extraction of' change to 'extraction of the' 7. 'the cloud and cloud shadows in this process' change to 'cloud and cloud shadow in this process' 8: What is the time period referred to by the statement 'over longer time periods' 10: remove 'if ' before equation

Page 7 5: 'will be' change to 'will also be' 8: 'values of' change to 'values for' 10: 'values of' change to 'values for' 12: 'extracted as' change to 'extracted as the'

Page 9 9: 'high consistency' change to 'highly consistent' 11: 'respectively (Figure 3).' change to 'respectively is shown in figure 3.'

Page 12 3: 'series surface' change to 'series of the surface' 4: 'area such as' change to 'area; including; 6: 'as a cross-validation' remove 'a' 8: 'models' change to 'model'

**Response:** All these grammatical errors have been corrected.

**Question:** Table 19: need to include uncertainty accessioned with the values given

**Response:** Because in the Section 4 we have already assessed the accuracy of the datasets. And we did not have real ground truth data whole the China, so we only calculated the surface water area in different regions directly.

**Question:** Page 13 Figure 5: There are no error bars on this figure – they need to be added or an explanation as to why they are not shown. The axis tick marks need to be 'out' rather than 'in' to improve clarity. The x axis labelling is cluttered and needs revision to make clearer

**Response:** We have tried to add the error bars in this figure, but when we added them on it, the figure looks very crowded and the useful information like the change points in different time will be buried. But we improved the clarity about the axis and the curves in each year based on the above suggestions.

**Questions:** Page 14 11: '2016 was' change to '2016 has been' 11: 'series and' change to 'series with' 12: 'of surface' change to 'of a surface' 12: 'in China' change to 'for China' 13:'in high consistency' change to 'is highly consistent' 14: 'data in' change to 'data from'16: '0.88 in' change to '0.88 for the' 18: 'and Poyang Lake region) with the GSW dataset, especially for the large water bodies (as lakes and' change to 'and Poyang Lake region) to that of the GSW data set, especially for large water bodies (such as lakes and' 19: 'and the' change to 'and' 20: remove 'for' 21: 'process' change to 'processes'

**Response:** All these grammatical errors have been corrected.

**Part 2 List of All Relevant Changes**

1. Grammatical modification of the full text.

2. Dataset and citation DOI update.

3. Added a new table of national and regional surface water related datasets of China in Section 1.

4. Refined the method section and updated Figure 1.

5. Improved the accuracy evaluation of time series data and added a new figure (Figure 4).

6. Expanded the section of 'Applications and data availability', updated Figure 5, and replaced the Figure 6.

7. In the part of discussion and conclusion, the uncertainty of data and methods and the application prospect of methods are added.

8. Adjusted the sort of funding information and supplemented some references.

**Part 3 Marked-up Manuscript**

[revised manuscript text omitted]

---

## Referee Report (RR1)

**Time series of Inland Surface Water Dataset in China (ISWDC) for 2000-2016 derived from MODIS archives**

Lu, S., Ma, J., Ma, X., Tang, H., Zhao, H., and Hasan Ali Baig, M., 2018:
Earth Syst. Sci. Data Discuss., https://doi.org/10.5194/essd-2018-134.

**Overview**

This paper describes the derivation of a dataset, the Inland Surface Water Data in China (ISWDC), from MODIS imagery. The dataset indicates the presence (as 1) or absence (0) of surface water in China for the period from 2000 to 2016 at a spatial resolution of 250 m and temporal resolution of 8 days.

The paper has been reviewed. The main points from the two reviews were:
1. The significant overlap with an earlier paper by the same authors, especially the methodology section (cited paper by Lu et al., 2017): Reviewer 1.
2. Simplistic and superficial treatment of uncertainties: Reviewers 1 & 2
3. Incorrect and/or general statements: Reviewer 1
4. The wider and long-term application of the approach: Reviewer 2

This review of the revised manuscript will indicate the extent to which these points have been addressed. In so doing, I take account of the key criteria for acceptance in this journal: (a) uniqueness; (b) usefulness and (c) completeness.

This is an improved version. I recommend a further revision to address the outstanding points raised in the Specific and Technical Comments below.

**Specific Comments**

1. Overlap

   This is now much reduced.

2. Simplistic and superficial treatment of uncertainties

   The author response to Reviewer 1 states "On the issue of accuracy evaluation, we have increased the accuracy analysis of temporal process data" and Reviewer 2 "In the new final section, we added a paragraph to discuss the overall uncertainties".

   There is certainly more material and discussion on the comparison with other datasets (national and the Global Surface Water (GSW) product) and on the uncertainties in the revised manuscript. However, this still needs further work.

   a. Some of the material in the author response should be included in the paper. For example in the response to Reviewer 1:

   "As the national land cover data in 2000, 2005, 2010 are based on 30 m Landsat images that mainly obtained in summer season. The water surface in these datasets can be equated with annual maximum water surface results. So we compared them with our maximum ISWDC of corresponding year. The calculated $R^2$ is based on the area of different size of water bodies. The larger the $R^2$, the better the consistency and the smaller the area error between the two

datasets. Furthermore, the results of confusion matrix are equivalent to pixel scale analysis although it's not as intuitive as visual contrast."

This makes it clearer why the maximum/summertime values are used, especially when there is a factor of 2 change in the total surface area during the course of the year (Figure 7).

**b.** The comparison of the ISWDC and GSW data products has been extended to include time series of the annual size of permanent water bodies in China between 2000 and 2015 (new Figure 4 on page 12). It is not clear if water bodies below an area of 0.0625 $km^2$ in the GWS product have been included or not. This needs clarification.

Further, as performed by Klein et al., 2017 (cited reference), an analysis could and should be made of the performance of the product for pure and mixed water pixels.

**c.** The paragraph in the final section on the overall uncertainties is simply a restatement of the results presented In Table 2.

Finally, there would be real merit in comparing the ISWDC dataset with the Global WaterPack product of Klein et al., 2017 (cited reference) over China, especially as they are both based on MODIS imagery.

3. Incorrect and/or general statements

These have been addressed in the author response to the review comments and in the revised manuscript.

4. The wider and long-term application of the approach

A paragraph has been added on the global applicability of the approach and its extension to other water body indices and other sensors (e.g., Sentinel-2).

**Data Availability**

The dataset is available from http://doi.org/10.5281/zenodo.2616035. The dataset comprises a large number of files in 2 archive files. Read me files are included.

**Technical Comments**

Page 3: A new Table 1 has been included here in the revised manuscript. All the following Tables (Table 1, Page 7; Table 2, Page 11 and Table 3, page 16) need to be renumbered and the references to the tables revised.

Page 4, line 13: "China is one of the countries that have the highest densities of rivers and lakes in the world" should be replaced by "China has one of the highest densities of rivers and lakes in the world".

Page 7, Table 1: The final entry for 2015 (28) is either out of place or incorrect. Please check and amend as needed.

Page 10, lines 6-8: The 4 occurrences of "lesser than" should be the original "less than".

Page 10, line 10: The "determinant coefficients ($R^2$)" should be the "coefficient of determination ($R^2$)"

Page 11: Table 2. The table second part of the caption ("the assess results") does not make sense.

Page 11, line 4: Replace "whole China" with "the whole of China". Also Page 12, caption of Figure 4.

Page 15, Figure 6: It is still hard to distinguish the individual years. The use of coloured lines might provide some discrimination.

Page 15, Figure 7: The label for the y-axis is not the "Average annual area". It is the "area of surface water" of each region for the particular 8-day period (or similar). The precision of the numbers on the y-axis needs to be increased (0,1,1,2,2,3,3,4) -> (0,0.5,1,1.5,2,2.5,3.3.5,4).

Page 19, line 18: The Gao reference is missing the journal (Remote Sensing of the Environment).

The paper could do with a careful proof read of the English as there are other non-English constructions, e.g., neglect of the definite (the) and indefinite article (a/an).

---

## Author Response (AR2)

**Response to the Topical Editor**

**Specific Comments**

**Question:** Some of the material in the author response should be included in the paper. For example in the response to Reviewer 1: "As the national land cover data in 2000, 2005, 2010 are based on 30 m Landsat images that mainly obtained in summer season. The water surface in these datasets can be equated with annual maximum water surface results. So we compared them with our maximum ISWDC of corresponding year. The calculated $R^2$ is based on the area of different size of water bodies. The larger the $R^2$, the better the consistency and the smaller the area error between the two datasets. Furthermore, the results of confusion matrix are equivalent to pixel scale analysis although it's not as intuitive as visual contrast."

This makes it clearer why the maximum/summertime values are used, especially when there is a factor of 2 change in the total surface area during the course of the year (Figure 7).

**Response:** We added this response in Section 4.1 as a new paragraph.

**Question:** The comparison of the ISWDC and GSW data products has been extended to include time series of the annual size of permanent water bodies in China between 2000 and 2015 (new Figure 4 on page 12). It is not clear if water bodies below an area of 0.0625 $km^2$ in the GWS product have been included or not. This needs clarification.

**Response:** The water bodies in the GWS product with an area less than 0.0625 $km^2$ are not included in the comparison. We added an explanation in the first sentence in section 4.2 in order to make this question clearer.

**Question:** Further, as performed by Klein et al., 2017 (cited reference), an analysis could and should be made of the performance of the product for pure and mixed water pixels.

**Response:** In our algorithm, if the water pixel is extracted according to the principle of maximum variance between classes and minimum variance intra-class, it is recognized as pure water pixel. But in theory the most peripheral pixel of the water body must be a mixed pixel. In our product, we did not do further decomposition. In the process of accuracy evaluation, all the pixels in the center of water body and the most peripheral mixed pixels are treated as pure water. The final result of accuracy evaluation is actually the synthesis of pure water and the marginal mixed pixels. If the pure water pixels are separately evaluated, the accuracy of the results will be higher. Although the accuracy of mixed pixel regions may be reduced, but due to the proportion of mixed pixel is actually very low, we do not evaluate the precision separately.

**Question:** The paragraph in the final section on the overall uncertainties is simply a restatement of the results presented in Table 2.

**Response:** We reorganized the three paragraphs and modified some expressions on the accuracy and uncertainty analysis of the results in the final section.

**Question:** Finally, there would be real merit in comparing the ISWDC dataset with the Global WaterPack product of Klein et al., 2017 (cited reference) over China, especially as they are both based on MODIS imagery.

**Response:** Yes, indeed. Since the data produced by Igor Klein's group is not publicly shared, before we started writing this article, we had contacted the authors to obtain the data in China so we can make comparison and evaluation between the two data sets. Unfortunately, because of the post adjustment of data processing personnel, they did not provide their data eventually, so our research failed to compare with their data. If their data are publicly published in future, we will make some comparisons again.

**Technical Comments**

**Question:** Page 3: A new Table 1 has been included here in the revised manuscript.

All the following Tables (Table 1, Page 7; Table 2, Page 11 and Table 3, page 16) need to be renumbered and the references to the tables revised.

**Response:** We renumbered the 4 tables and revised the references of the tables in the main body of the manuscript.

**Question:** Page 4, line 13: "China is one of the countries that have the highest densities of rivers and lakes in the world" should be replaced by "China has one of the highest densities of rivers and lakes in the world".

**Response:** We replaced the sentence of "China is one of the countries that have the highest densities of rivers and lakes in the world" by the suggested one.

**Question:** Page 7, Table 1: The final entry for 2015 (28) is either out of place or incorrect. Please check and amend as needed.

**Response:** We filled in the missing number. The correct number is 289.

**Question:** Page 10, lines 6-8: The 4 occurrences of "lesser than" should be the original "less than".

**Response:** We corrected these misused words.

**Question:** Page 10, line 10: The "determinant coefficients $(R^2)$" should be the "coefficient of determination $(R^2)$"

**Response:** We corrected this wrong expression.

**Question:** Page 11: Table 2. The table second part of the caption ("the assess results") does not make sense.

**Response:** We changed "the assess results" to "the accuracy evaluation results".

**Question:** Page 11, line 4: Replace "whole China" with "the whole of China". Also Page 12, caption of Figure 4.

**Response:** We replaced the two places with the "the whole of China"

**Question:** Page 15, Figure 6: It is still hard to distinguish the individual years. The use of coloured lines might provide some discrimination.

**Response:** We changed the lines and marks in different colors in this figure.

**Question:** Page 15, Figure 7: The label for the y-axis is not the "Average annual area". It is the "area of surface water" of each region for the particular 8-day period (or similar). The precision of the numbers on the y-axis needs to be increased (0,1,1,2,2,3,3,4) -> (0,0.5,1,1.5,2,2.5,3.3.5,4).

**Response:** We revised the label for the y-axis as "Area of surface water" and revised the caption as "The 8-day surface water area in different regions of China …..". Furthermore, the precision of numbers on the y-axis been changed as (0, 5, 10, 15, 20, ……).

**Question:** Page 19, line 18: The Gao reference is missing the journal (Remote Sensing of the Environment).

**Response:** We added the journal Name of "Remote Sensing of Environment".

**Question:** The paper could do with a careful proof read of the English as there are other non-English constructions, e.g., neglect of the definite (the) and indefinite article (a/an).

**Response:** We have carefully made a proof check of the english grammer.

[revised manuscript text omitted]